# Exploring Core Genes by Comparative Transcriptomics Analysis for Early Diagnosis, Prognosis, and Therapies of Colorectal Cancer

**DOI:** 10.3390/cancers15051369

**Published:** 2023-02-21

**Authors:** Md. Ariful Islam, Md. Bayazid Hossen, Md. Abu Horaira, Md. Alim Hossen, Md. Kaderi Kibria, Md. Selim Reza, Khanis Farhana Tuly, Md. Omar Faruqe, Firoz Kabir, Rashidul Alam Mahumud, Md. Nurul Haque Mollah

**Affiliations:** 1Bioinformatics Lab (Dry), Department of Statistics, University of Rajshahi, Rajshahi 6205, Bangladesh; 2Hubei Key Laboratory of Agricultural Bioinformatics, College of Informatics, Huazhong Agricultural University, Wuhan 430070, China; 3Department of Computer Science and Engineering, University of Rajshahi, Rajshahi 6205, Bangladesh; 4Department of Otorhinolaryngology-Head & Neck Surgery, School of Medicine, University of Maryland, Baltimore, MD 21201, USA; 5NHMRC Clinical Trials Centre, Faculty of Medicine and Health, The University of Sydney, Camperdown, NSW 2006, Australia

**Keywords:** colorectal cancer, gene expression profiles, core genes, early diagnosis, prognosis, therapies, integrated statistics and bioinformatics approaches

## Abstract

**Simple Summary:**

Colorectal cancer (CRC) is a complex disease that has a high mortality rate. This study explored CRC-related core genes (CGs) from multiple microarray gene-expression profiles in the NCBI-GEO database by combining some statistics and bioinformatics techniques. It also disclosed their molecular functions, biological processes, cellular components, signaling pathways, and transcriptional and post-transcriptional regulatory factors by using different online bioinformatics tools and databases. The prognostic power of CGs was investigated from the independent TCGA database by using survival probability curves and box plots of CGs-expressions in different stages (control, Stage 1, Stage 2, Stage 3, and Stage 4) of CRC. Finally, a few CGs-guided drug molecules were suggested for the treatment of CRC by molecular docking and dynamic simulation studies. Therefore, the findings of this study would be useful resources for early diagnosis, prognosis, and therapies of CRC.

**Abstract:**

Colorectal cancer (CRC) is one of the most common cancers with a high mortality rate. Early diagnosis and therapies for CRC may reduce the mortality rate. However, so far, no researchers have yet investigated core genes (CGs) rigorously for early diagnosis, prognosis, and therapies of CRC. Therefore, an attempt was made in this study to explore CRC-related CGs for early diagnosis, prognosis, and therapies. At first, we identified 252 common differentially expressed genes (cDEGs) between CRC and control samples based on three gene-expression datasets. Then, we identified ten cDEGs (*AURKA, TOP2A, CDK1, PTTG1, CDKN3, CDC20, MAD2L1, CKS2, MELK,* and *TPX2*) as the CGs, highlighting their mechanisms in CRC progression. The enrichment analysis of CGs with GO terms and KEGG pathways revealed some crucial biological processes, molecular functions, and signaling pathways that are associated with CRC progression. The survival probability curves and box-plot analyses with the expressions of CGs in different stages of CRC indicated their strong prognostic performance from the earlier stage of the disease. Then, we detected CGs-guided seven candidate drugs (Manzamine A, Cardidigin, Staurosporine, Sitosterol, Benzo[a]pyrene, Nocardiopsis sp., and Riccardin D) by molecular docking. Finally, the binding stability of four top-ranked complexes (TPX2 vs. Manzamine A, CDC20 vs. Cardidigin, MELK vs. Staurosporine, and CDK1 vs. Riccardin D) was investigated by using 100 ns molecular dynamics simulation studies, and their stable performance was observed. Therefore, the output of this study may play a vital role in developing a proper treatment plan at the earlier stages of CRC.

## 1. Introduction

Cancer is a complex disease caused by multiple alterations at the genetic and epigenetic levels that increasingly lead to abnormal cell division and cellular transformation [1,2]. Colorectal cancer (CRC) is the third most common solid malignancy and the second deadliest tumor worldwide [3]. The CRC incidence is expected to rise by 60%, with 2.2 million new cases and 1.1 million deaths globally by 2030 [4]. The number of new incidences and mortalities is increasing due to insufficient evidence about diagnostic biomarkers and the molecular mechanism of CRC [4]. Early detection of CRC is associated with lower morbidity and mortality and a higher survival rate compared with late detection. For example, the five-year survival rate increases from 11% (late detection) to 90% at early detection of CRC [5]. However, the survival rate significantly decreases, and the cost of treatment increases, whereas CRC is identified in later stages compared to earlier stages [3,6,7]. Therefore, early diagnosis, prognosis, and therapies leads to reduce CRC-related mortality [7]. Several factors, including excessive alcohol consumption, obesity, unhealthy dietary habits, and an abnormal lifestyle, are all considered non-causal risk factors for CRC development. However, these non-causal risk factors cannot be used for CRC detection at an earlier stage.

Generally, differentially expressed genes (DEGs) between cancer and control samples are considered as the cancer-causing/stimulating genes. A gene may show a differential expression pattern between cancer and control samples for several reasons, including mutation, DNA methylation, and other epigenetic stimulations. The genes that are associated with the development of cancer, are known as oncogenes (upregulated DEGs) and tumour-suppressor genes (down-regulated DEGs) [1,2,8]. Thus, cancer incidence, development, recurrence, and non-recurrence are associated with pathogenetic processes of DEGs [9]. Several studies reported some dysregulating genes in the CRC cases compared to non- CRC cases that are associated with CRC proliferation, differentiation, apoptosis, metastasis, recurrence, and lower survival [10,11,12,13]. Many earlier transcriptomics studies explored the pathogenetic processes of CRC through DEGs [14,15,16,17,18,19,20,21]. However, none of them discussed rigorously about early diagnosis, prognosis, and therapies for CRC. Patil et al. (2021) [22] identified forty CRC-causing/stimulating core DEGs and recommended their application for the early diagnosis of CRC. Though the number 40 is much smaller than the whole genome size, it may also not be suitable for further investigation by the wet-lab researchers, since wet-lab experiments with 40 DEGs might be costly, time-consuming, and laborious. So, a smaller set of core DEGs might be required for further experimental investigation. On the other hand, this study did not provide any recommendations about suggested core DEGs guide any drug molecules for therapies for CRC. Therefore, the present study attempted to discover CRC-causing/stimulating core-DEGs, highlighting their pathogenetic processes for early prognosis, diagnosis, and therapies of CRC. The pipeline of this study is given in Figure 1.

## 2. Materials and Methods

### 2.1. Data Sources and Descriptions

The necessary datasets that were analyzed in this study are described below.

#### 2.1.1. Collection of Microarray Datasets to Explore CRC-Causing Core Genes

We downloaded three microarray datasets of CRC using the accession IDs GSE106582, GSE110223, and GSE74602 from the NCBI Gene Expression Omnibus (GEO) database. The GPL10558 platform was used for the GSE106582 dataset, which contains 194 samples, including 77 cancer and 117 adjacent tissue samples. The GPL96 platform was used for the GSE110223 dataset, which contains 26 samples, including 13 cancer and 13 adjacent tissue samples. The GPL6104 platform was used for the GSE74602 dataset, which contains 60 samples, including 30 cancer and 30 adjacent tissue samples.

#### 2.1.2. Collection of Drug Molecules Set for Drug Repositioning

For identifying candidate repurposed drugs, we collected target receptor-guided drug molecules from the DSigDB [23] database (Appendix A).

### 2.2. Method for Identification of DEGs

We used the GEO2R web tool [24] based on the LIMMA (linear models for microarray data) approach to identify DEGs between cancer and adjacent tissue samples for each of the three datasets. The LIMMA method uses modified *t*-statistics to calculate *p*-values. We used the Benjamini–Hochberg (BH) approach [25] to adjust the *p*-values. The log_2_ fold-change (Log_2_FC) and adjusted *p*-values were used to separate the up- and down-regulated DEGs by the following cut-offs:(1)$$DEGs_{g}=\left\{\begin{array}{c} \mathrm{DEG}\;\left(\mathrm{Upregulated}\right),\;\;\;\;\;\;\;\;\;\;\;\;\;\;\;\;\mathrm{if}\;\;\;\mathrm{adj}.p.\mathrm{value}<0.05\;\mathrm{and}\;{\mathrm{Log}}_{2}{\mathrm{FC}}_{g}>+1.0\\ \mathrm{DEG}\;\left(\mathrm{Downregulated}\right),\;\;\;\;\;\;\;\;\;\;\;\mathrm{if}\;\;\;\mathrm{adj}.p.\mathrm{value}<0.05\;\mathrm{and}\;{\mathrm{Log}}_{2}{\mathrm{FC}}_{g}<-1.0 \end{array}\right.$$

We considered common DEGs (cDEGs) from three datasets to identify core CGs. All DEGs were visualized using a Venn diagram using FunRich 3.1.3 [26].

### 2.3. Protein-Protein Interaction (PPI) Network Analysis

The protein-protein interaction (PPI) network was utilized to detect core genes (CGs) from cDEGs. We considered the STRING database [27] with a median confidence score (MCS) of 0.4 to produce a PPI network of cDEGs and Cytoscape software for better visualization of the network [28]. The CGs were selected from the PPI network using the CytoHubba plugin in Cytoscape [28,29]. The present study used maximal clique centrality (MCC) topology analysis methods to identify the CGs.

### 2.4. Association of CGs with Different Stages of CRC Progression

To investigate the association of CGs with the different stages of CRC based on independent databases, we performed box-plot analysis based on their expression levels in different CRC progression stages (Normal status, Stage 1, Stage 2, Stage 3, and Stage 4) through the UALCAN web tool with the TCGA-COAD and TCGA-READ databases [30,31].

### 2.5. Prognosis Power of CGs

To investigate the prognosis power of CGs by multivariate Kaplan–Meier survival probability curves, we considered the SurvExpress web tool based on the TCGA-COAD and TCGA-READ databases (https://portal.gdc.cancer.gov/exploration, accessed date: 2 January 2022) [30,32]. The log-rank test was used in SurvExpress, and the risk group hazard ratio with a 95% confidence interval was included in the Kaplan-Meier survival plot [32]. The *p*-value < 0.05 was used as the cut-off.

### 2.6. CGs-Set Enrichment Analysis

CGs-set enrichment analysis (cGSEA) determines the classes of genes or proteins that are over-represented (enriched) in a predefined large set of genes or proteins that are associated with the terms of interest, including gene ontology (GO), pathways, diseases, chemicals, drugs, biomolecules (miRNA, TFs), and so on. To detect the significantly enriched terms of interest, let *A_i_* be the predefined gene-set in the *i*th term of interest, and *M_i_* denotes the number of CGs in *A_i_* (*i* = 1, 2,…, *r*); *T* is considered as an enriched gene number that created a combined set A such that $A=\cup_{i=1}^{r}A_{i}=A_{i}\cup A_{i}^{c}$ and $T\leq \overset{r}{\underset{i=1}{\sum }}M_{i};$ where $A_{i}^{c}$ is considered as the complement-set of *A_i_*. Again, suppose *t* represents the number of CGs and *m_i_* denotes the number of CGs subset of *A_i_*. Table 1 summarizes these results.

The Enrichr web tool [33] was considered to investigate the association of CGs with terms of interest. This web tool uses the Fisher exact test to examine the significance of the association between CGs and *i*th term of interest.

### 2.7. Association of CGs with Different Diseases

We considered the Enrichr web tool [33] to verify the association of CGs with different diseases, including CRC, using the DisGeNET database, which was constructed based on 21,671 genes and 30,170 diseases [34]. It measures the association of a disease with a group of CGs that are overlapped (common) with the reference gene set of that disease (see Table 1). To investigate the pan-cancer role of CGs, we performed the pan-cancer analysis of each CG by the TIMER 2.0 web tool [35] with the TCGA database [36]. In both cases, the *p*-value < 0.05 was selected as the cut-off for statistical significance.

### 2.8. Association of CGs with GO Terms and KEGG Pathway

To disclose the pathogenetic processes of CGs, we performed CGs-set enrichment analysis with GO terms and pathways by using the Enrichr web tool [33]. Biological process (BPs), molecular function (MFs), and cellular component (CCs) were investigated to explore potential GO terms and pathways based on the KEGG database as displayed in Table 1. A *p*-value < 0.001 was used as the cut-off for the statistical significance.

### 2.9. CGs Regulatory Network Analysis

A gene regulatory network (GRN) provides information about molecular regulators that connect to regulate the gene expression level of mRNA. Transcription factors (TFs) and microRNAs (miRNAs) are considered the major regulators of gene expression. TFs proteins are regarded as the significant contributors to GRN because they bind to a particular region of DNA (enricher/promoter) and influence gene expression at the transcriptional level. A miRNA is a non-coding RNA considered a central post-transcriptional regulator of gene expression. The human genome contains up to 1600 TFs and 1900 miRNAs. A TFs vs. CGs network is considered an undirected graph, where nodes represent TFs or CGs and edges depict interactions between TFs and CGs, respectively. A TF-node is considered the major regulatory factor for CGs if it contains the largest number of interactions with CG nodes. We considered regulatory analysis of CGs (transcription factors (TFs) vs. CGs and micro-RNAs (miRNAs) vs. CGs) through Network Analyst [37] platform-based JASPAR [38] and TarBase [39] databases, respectively, to detect the core transcriptional and post-transcriptional regulators of CGs. For better illustration, we used Cytoscape software [28]. The core regulators were chosen by utilizing degree [40] and betweenness [41] scores.

### 2.10. Molecular Docking

We conducted molecular docking studies of receptors and drug molecules to explore FDA-approved repurposable drugs for CRC. CGs-mediated proteins and related TFs proteins were considered drug target receptors (*p* = 14). The online database DSigDB was used to extract CGs-guided drug agents. The 3-dimensional (3D) structures of AURKA, TOP2A, CDK1, PTTG1, CDC20, MAD2L1, CKS2, MELK, TPX2, YY1, and SRF targets were retrieved from the Protein Data Bank (PDB) [42] with IDs 6VPM, 5NNE, 5LQF, 7NJ1, 4GGC, 2V64, 5LQF, 5M5A, 6VPM, 4C5I, and 1HBX, respectively. The remaining targets, FOXC1, CDKN3, and NFIC, were retrieved from the SWISS-MODEL [43] with the Uniport IDs Q12948, P08651, and Q16667, respectively. Using the PubChem database [44], we retrieve the 3D structures of all (*q* = 158) drug molecules. The visualization of the receptor proteins and co-crystal ligands were performed via the Discovery Studio Visualizer 2019 [27]. The receptor proteins were processed using AutoDock tools [45] and the Swiss PDB viewer by adding the structural charges and reducing the energy of receptors, respectively [45,46]. The docked complexes were analyzed through Discovery Studio Visualizer 2019. Let *B_ij_* be the binding affinity (BA) score of *i*th receptors (*i* = 1, 2,…, *p*) and *j*th drugs (*j* = 1, 2,..., *q*). The receptors and drug molecules were sorted by the decreasing order of their average BA score for selecting the top-ordered few potential candidate repurposable drugs.

### 2.11. Molecular Dynamics (MD) Simulation

We performed MD simulations of the top-orderedprotein–ligand complexes (TPX2–Manzamine A, CDC20–Cardidigin, MELK–Staurosporine, and CDK1–Riccardin D) through the YASARA software (Version: 22.8.22) [47] based on the AMBER14 force field [48]. Prior to simulation, the hydrogen-bonding network of each complex in a simulated cell was optimized using a TIP3P water model [45]. The periodic limit conditions were kept constant at 0.997 gL-1 of solvent concentration. The primary energy was minimized in each simulation by considering the steepest gradient technique with 5000 cycles. The complexes “TPX2–Manzamine A”, “CDC20–Cardidigin”, “MELK–Staurosporine”, and “CDK1–Riccardin D” consist of a total of 56,287, 35,859, 81,347, and 45,153 atoms, respectively. At the Berendsen thermostat [49] and constant pressure, a 100 ns MD simulation was examined. Please see our previous publications for details about the MD simulation strategy [50,51,52,53]. For subsequent analysis, we took snapshots of the trajectories every 250 ps, ran them via the built-in script of YASARA [54] macro, and calculated the binding free energy of the MM-Poisson–Boltzmann surface area (MM-PBSA) by analyzing all the snapshots [55]. To calculate binding-free energy, we used the following formula:(2)$$\mathrm{Binding}\;\mathrm{free}\;\mathrm{energy}=E_{\mathrm{potReceptor}}+E_{\mathrm{solvReceptor}}+E_{\mathrm{potLigand}}+E_{\mathrm{potComplex}}-E_{\mathrm{solvComplex}}$$

More positive binding energy represents stronger binding.

## 3. Results

### 3.1. Identification of DEGs

To identify DEGs from each of three microarray gene-expression datasets, we used the statistical LIMMA approach through the GEO2R web tool, with the cut-off at adjusted *p*-value < 0.05 and |log_2_(fold change)| > 1. In the GSE106582 dataset, we identified 594 DEGs, including 213 upregulated and 381 downregulated genes. In the GSE110223 dataset, we identified 625 DEGs that contain 260 upregulated and 365 downregulated genes. In the GSE74602 dataset, we identified 1674 DEGs, including 673 upregulated and 1001 downregulated genes. The Venn diagram in Figure 2 visualizes the common DEGs among the three datasets. The Venn diagram exhibits 252 cDEGs among the three datasets.

### 3.2. Identification of Core Genes (CGs)

We construct the PPI network of cDEGs and visualize the PPI network to identify the potential genes most significantly associated with the development of CRC. The PPI network contains 216 nodes and 616 edges, with a confidence score of 0.40. Then, the MCC topology analysis method of CytoHubba was performed to calculate the top-ranked CGs within the network. We found the ten top-ranked (*AURKA, TOP2A, CDK1, PTTG1, CDKN3, CDC20, MAD2L1, CKS2, MELK,* and *TPX2*) genes (see Figure 3). These top ten CGs were identified as major controllers of CRC and considered for subsequent analysis.

### 3.3. Association of CGs with Different Stages of CRC Progression

The box-plot analysis based on an independent database represents a high difference between normal expression and every CRC progression stage (Stage 1, Stage 2, Stage 3, and Stage 4) expression of all CGs (see Figure 4A and Appendix A). So, our proposed CGs have strong prognostic power to identify CRC at an earlier development stage.

### 3.4. Prognosis Power of CGs

A survival analysis was performed to examine the prognosis power of CGs. The multivariate Kaplan–Meier survival plot of CGs expressions using the TCGA-COAD and TCGA-READ databases represents a significant difference (*p*-value < 0.001) between lower-risk and higher-risk groups (see Figure 4B).

### 3.5. Association of CGs with Different Diseases

The enrichment analysis of the CGs-set with different diseases based on the DisGeNET database showed that the CGs-set is significantly associated with various diseases (*p*-value < 0.05). Figure 3A and Appendix A show the top-ranked 20 diseases, all of which are different types of cancer, including CRC. We observed that 3 CGs (*MELK, CKS2, CDC20*) and 2 CGs (*MELK, CDC20*) do not overlap with the reference gene-sets of colon carcinoma and colorectal carcinoma, respectively (Figure 5A and Appendix A, and Appendix A). These results suggested a pan-cancer role for CGs (Appendix A). To investigate the pan-cancer role of CGs, we also performed pan-cancer analysis based on the TCGA database. We selected the top-ranked 20 cancers as displayed in Figure 5B. Figure 5A,B commonly showed that eight cancers (colon adenocarcinoma, bladder urothelial carcinoma, esophageal carcinoma, glioblastoma multiforme, liver hepatocellular carcinoma, lung adenocarcinoma, prostate adenocarcinoma, and stomach adenocarcinoma) are significantly associated with CGs.

### 3.6. Association of CGs with GO Terms and KEGG Pathway

Enrichment analysis of the CGs was performed using the Enrichr web tool. Table 2 shows the annotated GO terms in three categories (BPs, CCs, and MFs). In the case of biological processes (BPs), CGs were mainly involved in mitotic cell-cycle phase transition (GO:0044772), anaphase-promoting complex-dependent catabolic process (GO:0031145), regulation of G2/M transition of mitotic cell cycle (GO:0010389), mitotic spindle organization (GO:0007052), regulation of mitotic cell cycle (GO:0007346). In molecular function (MFs), CGs were mainly involved in histone kinase activity (GO:0035173), RNA polymerase II CTD heptapeptide repeat kinase activity (GO:0008353), protein kinase binding (GO:0019901), CXCR chemokine receptor binding (GO:0045236), cyclin-dependent protein kinase activity (GO:0097472), etc. In cellular components (CCs), CGs were mainly involved in the spindle (GO:0005819), cyclin-dependent protein kinase holoenzyme complex (GO:0000307), serine/threonine-protein kinase complex (GO:1902554), intracellular non-membrane-bounded organelle (GO:0043232), mitotic spindle (GO:0072686), etc. The KEGG pathway enrichment analysis results for CGs were also shown in Table 2. The KEGG pathways of CGs were enriched in the cell cycle, bladder cancer, oocyte meiosis, human T-cell leukemia virus one infection, progesterone-mediated oocyte maturation, etc.

### 3.7. Identification of Regulatory Factors

TFs proteins and miRNAs play a fundamental role in the modification of gene expression at the transcriptional and post-transcriptional levels, respectively. To explore the major transcriptional regulatory factors of CGs, we constructed a TFs vs. CGs interaction network where round nodes with red color represent the CGs and square nodes with green/purple color represent the TFs (see Figure 6A). TFs proteins vs. CGs regulatory analysis revealed four highest-ranking significant candidate TFs modifiers (NFIC, FOXC1, YY1, and GATA2) that may regulate the expression of CGs at the transcriptional level (see Figure 6A). Similarly, we constructed an undirected interaction network of miRNAs vs. CGs to reveal the post-transcriptional regulator of CGs, where red color nodes represent the CGs and green/blue color nodes illustrate the miRNAs (see Figure 6B). The miRNAs vs. CGs regulatory network analysis revealed six highly interacted non-coding RNAs (miRNAs) such as hsa-mir-147a, hsa-mir-129-2-3p, hsa-mir-124-3p, hsa-mir-34a-5p, hsa-mir-23b-3p, and hsa-mir-16-5p that act as gene expression regulators at the post-transcriptional level (see Figure 6B). So, those identified TFs and miRNAs may influence the gene expression of CGs at the transcriptional and post-transcriptional levels, respectively.

### 3.8. Drug Repurposing through Molecular Docking Studies

We considered 10 CG-guided proteins and 4 TFs proteins as target-receptor proteins for molecular docking. A structural interaction was carried out between target-receptor proteins and 158 drug agents by molecular docking studies, which computed the receptor-drug binding affinities (BA) for each interaction (see Figure 7).

Manzamine A, Cardidigin, Staurosporine, Sitosterol, Benzo[a]pyrene, Nocardiopsis sp., and Riccardin D were shown to have strong BA against all of the target receptors, and their average BA lies between −9.20 and −8.2 (kcal mol^−1^). Among those drugs, Manzamine A was shown to have the highest BA against almost every target protein, with an average BA of −9.2 kcal mol^−1^. Therefore, we proposed those seven drugs (Manzamine A, Cardidigin, Staurosporine, Sitosterol, Benzo[a]pyrene, Nocardiopsis sp., and Riccardin D) as candidate drug agents and displayed them in red color in Figure 7. We also revealed the structural interaction profiles of four top-ordered receptor proteins and drug complexes in Table 3.

### 3.9. Molecular Dynamic (MD) Simulations

TPX2–Manzamine A, CDC20–Cardidigin, MELK–Staurosporine, and CDK1–Riccardin D complexes have shown the highest BA in molecular docking analysis (Table 3). So, we considered those complexes for examining their binding stability through MD simulations. We observed that each protein-ligand complex (TPX2–Manzamine A, CDC20–Cardidigin, MELK–Staurosporine, and CDK1–Riccardin D) showed significant stability in a 100 ns MM-PSSA simulation (see Figure 8A). RMSD values corresponding to each complex were calculated (see Figure 8A). RMSD values showed lower flexibility around 1.5 Å to 3.0 Å for all four complexes. The average RMSD values for TPX2-Manzamine A, CDC20–Cardidigin, MELK–Staurosporine, and CDK1–Riccardin D complexes were 2.592 Å, 1.724 Å, 2.235 Å, and 2.516 Å, respectively. The CDC20–Cardigin complex showed a more substantial structural rigidity than the other three complexes, gained equilibrium at four ns, and displayed good stability after that. MELK–Staurosporine showed a gradual increase in RMSD before 22 ns, and after this time point, the RMSD score of the complex illustrated almost stable movement between 2.2 Å and 2.50 Å for 68 ns. After that, there were irregular fluctuations in the RMSD. On the contrary, CDK1-Riccardin D complexes exhibited instability, and the RMSD displayed an upward trend from 2.0 Å to 3.2 Å over time. Similarly, TPX2–Manzamine A showed irregular oscillation in RMSD between 1.7 Å and 3.3 Å. In addition, the MM-PBSA binding energy for four complexes was also computed. Figure 8B illustrates the binding energies of the complexes. On average, TPX2–Manzamine A, CDC20–Cardidigin, MELK–Staurosporine, and CDK1–Riccardin D complexes produced MM-PBSA binding energies of 84.39 KJ mol^−1^, −95.07 KJ mol^−1^, −235.86 KJ mol^−1^, and 154.39 KJ mol^−1^, respectively.

## 4. Discussion

To explore core genomic biomarkers and their mechanisms in the CRC progression, firstly, we identified 252 cDEGs between CRC and control samples, out of around 40,000 genes in three gene expression datasets of the NCBI-GEO database with accession numbers GSE106582, GSE110223, and GSE74602. Though the number of DEGs is much smaller than the total number of genes, it may still be a large number for further investigation by wet-lab experiments since it would be laborious, time-consuming, and costly. Therefore, a smaller set of DEGs that are representative of all DEGs is required to reduce time, cost, and labor during further experiments by the wet-lab researchers. Though total DEGs are more informative than any smaller set of DEGs, a smaller representative set of DEGs would be more beneficial from the viewpoints of time, cost, and labor. Therefore, in this study, we proposed 10 top-ranked DEGs (*AURKA, TOP2A, CDK1, PTTG1, CDKN3, CDC20, MAD2L1, CKS2, MELK,* and *TPX2*) as the core genes (CGs) for early diagnosis, prognosis, and therapies of CRC. The survival probability curves and box-plot analyses with the expressions of CGs in different stages (control, Stage 1, Stage 2, Stage 3, and Stage 4) of CRC with the TCGA database indicated their strong prognostic performance from the earlier stages of the disease. It should be mentioned here that Patil et al. (2021) [22] identified CRC-causing 40 CGs for early diagnosis, which might be a large number for further investigation by the wet-lab researchers since it would be laborious, time-consuming, and costly, as mentioned earlier. In our proposed 10 CGs, 9 genes (*AURKA, TOP2A, CDK1, PTTG1, CDKN3, CDC20, MAD2L1, MELK,* and *TPX2*) were overlapped/common with their 40 CGs. In addition, we also investigated the association of our proposed CGs with different diseases. We found they have a strong pan-cancer role, including CRC. The literature review also supported the association of our proposed CGs with CRC. For example, *AURKA* is considered an oncogene that significantly impacts the proliferation and progression of colorectal carcinoma from colorectal adenoma [56]. Generally, *AURKA* is overexpressed and amplified in CRCs [57,58,59,60,61,62]. *TOP2A* is highly expressed during tumor development and responds to drug therapy for CRC [63]. *CDK1* is also overexpressed and sensitive to apoptosis in CRC cells [64]. *CDKN3* is highly expressed in CRC tissues and remarkably related to patients’ diagnoses [65]. *CKS2* overexpression is correlated with aggressive tumor development in CRC, meaning that *CKS2* might function as a decent CRC biomarker [66]. *CKS2* is a promising biomarker contributing to CRC tumor development [66]. *CDC20*, *PTTG1*, and *MAD2L1* might be CRC stage-related genes [67]. *MELK* might play a role as an effective therapeutic target for CRC [68]. *TPX2* is highly upregulated in CRC tissues [69].

We identified the top-ranked five GO terms and KEGG pathways of CGs to reveal their molecular mechanisms in CRC progression. The identified GO term ‘cell cycle’ is one of the most important biological processes in the human body [70]. It has four sequential phases. Arguably, the most important phases are the S phase (DNA replication occurs) and the M phase (cell divides into two daughter cells) [71]. The fundamental task of the cell cycle is to ensure that DNA is faithfully replicated once during S phase and that identical chromosomal copies are distributed equally to two daughter cells during M phase. Usually, in adult tissue, there is a delicate balance between cell death and proliferation (cell division), producing a steady state. Disruption of this equilibrium by loss of cell cycle control may eventually lead to tumor development [72], including colorectal cancer [73], colon cancer [74], liver cancer [75], glioblastoma [76], breast cancer [77,78], lung cancer [79,80], gastric cancer [81], etc. So, the cell cycle is considered a vital cancer progression process. TOP2A is related to tumor development and poor survival outcomes by regulating cell proliferation and the CRC cell cycle [82,83]. CDK1 controls the cell cycle and aids in the development of colorectal tumors via an iron-regulated signalling axis [64]. In most CRCs, chromosomal variability resulting in an abnormal chromosome number, aneuploidy, was systematically related to a mitotic checkpoint’s loss of function [84,85]. The GO term ‘mitotic spindle orientation’ can influence tissue organization and control the placement of daughter cells within a tissue. Spindle misorientation greatly affects cancer development and progression [86], including CRC [87]. The top-ranked five MFs (cyclin-dependent protein kinase activity, CXCR chemokine receptor binding, protein kinase binding, RNA polymerase II CTD heptapeptide repeat kinase activity, and histone kinase activity) play a vital role in CRC development and proliferation [12,88,89,90]. Similarly, the enriched five CCs, including the spindle, mitotic spindle, cyclin-dependent protein kinase holoenzyme complex, intracellular non-membrane-bounded organelle, and serine/threonine-protein kinase complex, are strongly related to the progression of CRC [91,92,93,94,95]. Chromosomal instability happens in 80%–85% of CRCs and is considered the most common subtype of CRC [94]. Subsequent research showed that chromosomal instability is caused by mutations in the genes that govern the mitotic spindle checkpoint [93]. The consecutive activation of a group of serine-threonine kinases controls eukaryotic cell-cycle checkpoints [95]. We identified the top 5 enriched common KEGG pathways (cell cycle, bladder cancer [96], oocyte meiosis [22], human T-cell leukemia virus 1 infection, progesterone-mediated oocyte maturation) that are also reported by some other studies [22,96,97].

We also identified key regulators of CGs, such as four TFs (NFIC, FOXC1, YY1, and GATA2) and six miRNAs that played a significant role in CRC development. FOXC1, a member of the forkhead box family, has been connected to the growth and progression of numerous diseases [98,99], particularly CRC [100,101]. YY1 is a multipurpose TF protein that can stimulate or suppress gene expression [102] and plays a significant role in CRC tumor growth [103]. In CRC, the high-level expressions of GATA2 were linked with a poor prognosis and recurrence in solid tumors [104]. Nuclear factor 1 C-type (NFIC) regulates PFKB3 in response to CRC [105].

To find effective repurposable drugs against CRC, we performed molecular docking and computed binding scores among 158 CGs-associated drug agents and CGs-guided receptors. Then we proposed seven top-ordered drugs (Manzamine A, Cardidigin, Staurosporine, Benzo[a]pyren, Sitosterol, Nocardiopsis sp., and Riccardin D) based on binding affinities as the candidate repurposable drug. Manzamine A [106], Cardidigin [107], Staurosporine [108], Benzo[a]pyrene [109], Sitosterol [110], Nocardiopsis sp. [111], and Riccardin D [112] also suggested by some other studies for the treatment of CRC. Finally, the binding stability of top-docked four complexes (TPX2 vs. Manzamine A, CDC20 vs. Cardidigin, MELK vs. Staurosporine, and CDK1 vs. Riccardin D) was investigated by molecular dynamics (MD)-based MM-PBSA simulation and found their performance to be stable [113,114]. Thus, our findings may play a vital role in early diagnosis, prognosis, and therapies for CRC.

## 5. Conclusions

The present study identified the 10 top-ranked DEGs *(AURKA, TOP2A, CDK1, PTTG1, CDKN3, CDC20, MAD2L1, CKS2, MELK,* and *TPX2*) as the core genes (CGs), which showed Strong prognostic performance in the earlier stages of CRC. The CGs-disease and pan-cancer analysis commonly showed that CGs have a robust pan-cancer role, including CRC, bladder urothelial carcinoma, esophageal carcinoma, glioblastoma multiforme, liver hepatocellular carcinoma, lung adenocarcinoma, prostate adenocarcinoma, and stomach adenocarcinoma. The CGs regulatory network analysis detected some essential TFs proteins (NFIC, FOXC1, YY1, and GATA2) and miRNAs (hsa-mir-147a, hsa-mir-129-2-3p, hsa-mir-124-3p, hsa-mir-34a-5p, hsa-mir-23b-3p, and hsa-mir-16-5p) as the transcriptional and post-transcriptional regulators of CGs. The enrichment analysis also revealed some important CRC-causing GO terms and signaling pathways. For example, cell cycle and mitotic spindle pathways have a significant association with CRC progression. Finally, we recommended our proposed CGs-guided 7 candidate drug molecules (Manzamine A, Cardidigin, Staurosporine, Sitosterol, Benzo[a]pyrene, Nocardiopsis sp., and Riccardin D) for the treatment against the CRC by molecular docking analysis. Thus, the findings of this study may be more useful compared to the previous computation study results for early diagnosis, prognosis, and therapies forf CRC.

## Figures and Tables

**Figure 1 cancers-15-01369-f001:**
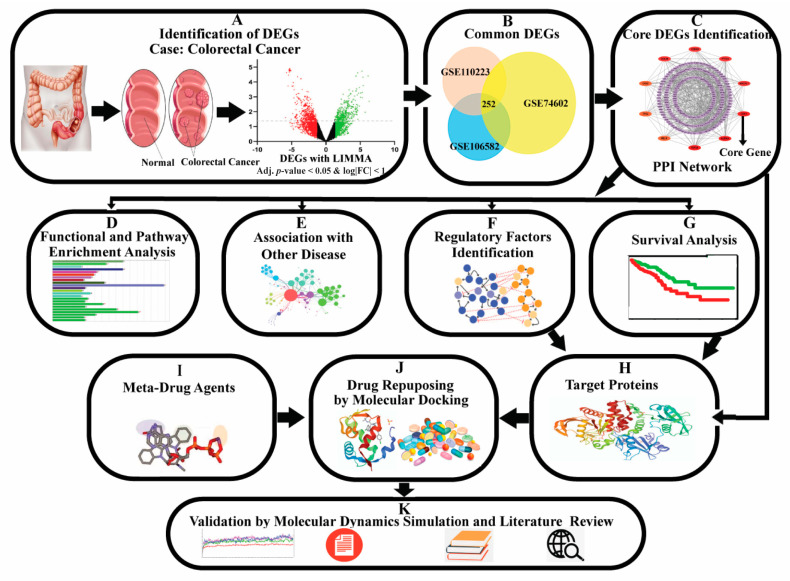
The pipeline of this study.

**Figure 2 cancers-15-01369-f002:**
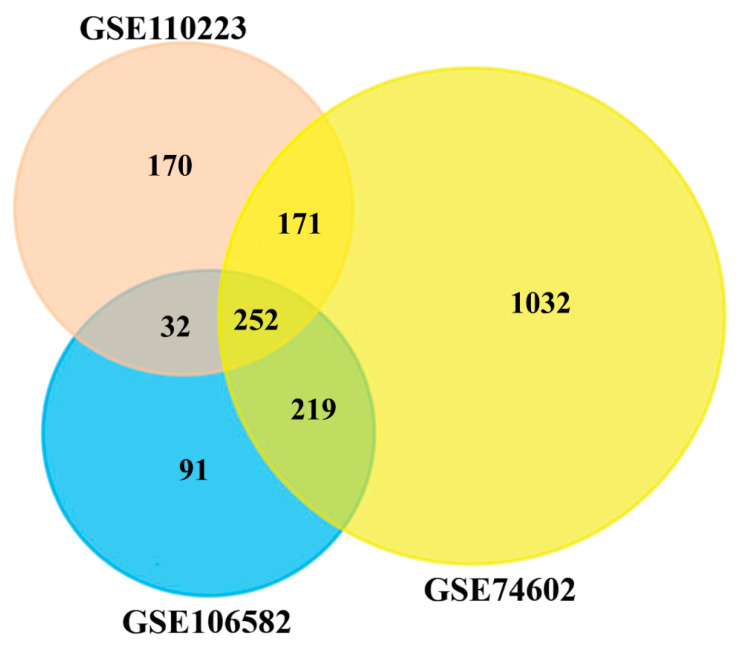
Common differentially expressed genes among GSE110223, GSE106582, and GSE74602 datasets were visualized through a Venn diagram, and 252 genes were found as cDEGs from CRC patients.

**Figure 3 cancers-15-01369-f003:**
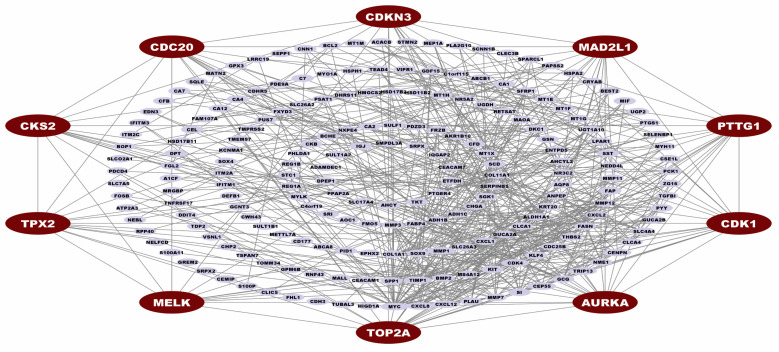
Protein-protein interaction network for cDEGs. Edges specify the interconnection between two proteins. The PPI network illustrates 216 nodes and 616 edges. Red color nodes (*AURKA*, *TOP2A*, *CDK1*, *PTTG1*, *CDKN3*, *CDC20*, *MAD2L1*, *CKS2*, *MELK*, and *TPX2*) represented the CGs.

**Figure 4 cancers-15-01369-f004:**
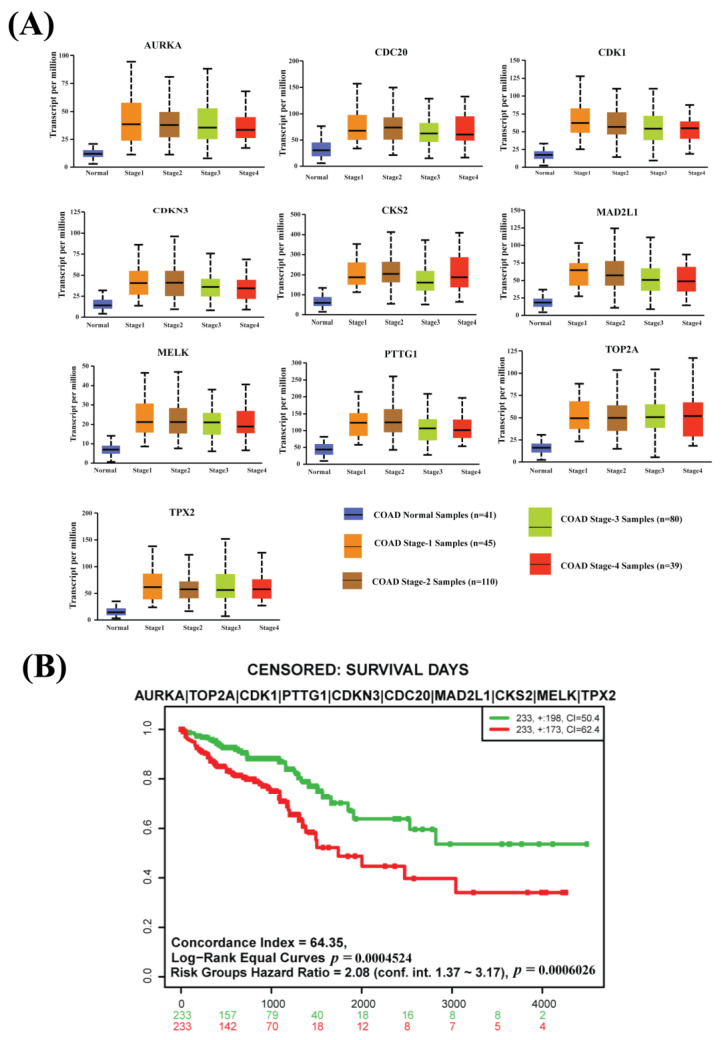
(**A**) Box plots for the expressions of CGs with various stages of colon adenocarcinoma (Stage 1, Stage 2, Stage 3, and Stage 4) using the COAD database and comparison with the control stage from the TCGA database. (**B**) The multivariate Kaplan–Meier survival probability plot of CRC patients with the CGs-expressions using the TCGA-COAD and TCGA-READ databases.

**Figure 5 cancers-15-01369-f005:**
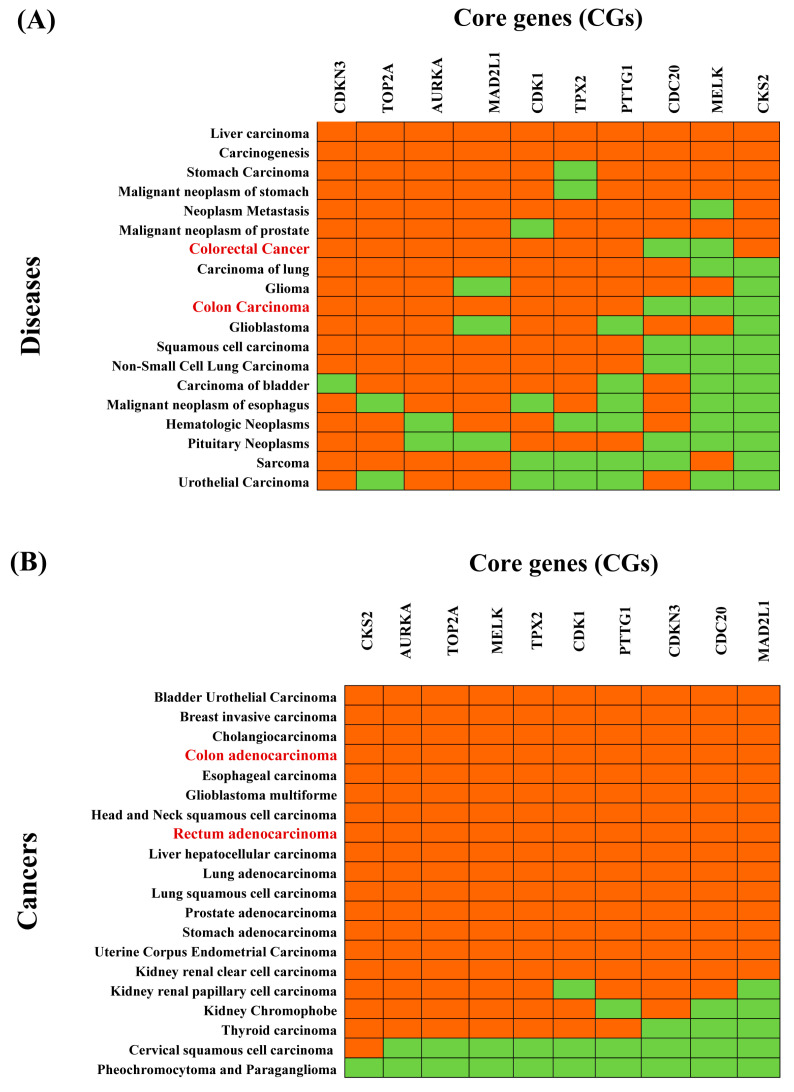
Association of CRC-causing CGs with different diseases. (**A**) Top 20 diseases associated with CGS obtained by disease-CGs association studies based on the Enrichr web tool with the DisGeNET database, where red and green colors indicate presence and absence, respectively. The red colour text in the column represent CRC-related cancers. (**B**) Top 20 cancers associated with CGs obtained by pan-cancer analysis based on the TIMER2 web tool with the TCGA database, where red and green colors indicate significant (*p*-value < 0.05) and insignificant (*p*-value ≥ 0.05) pairwise gene-disease association, respectively. The red colors text in the column represents CRC-related cancers.

**Figure 6 cancers-15-01369-f006:**
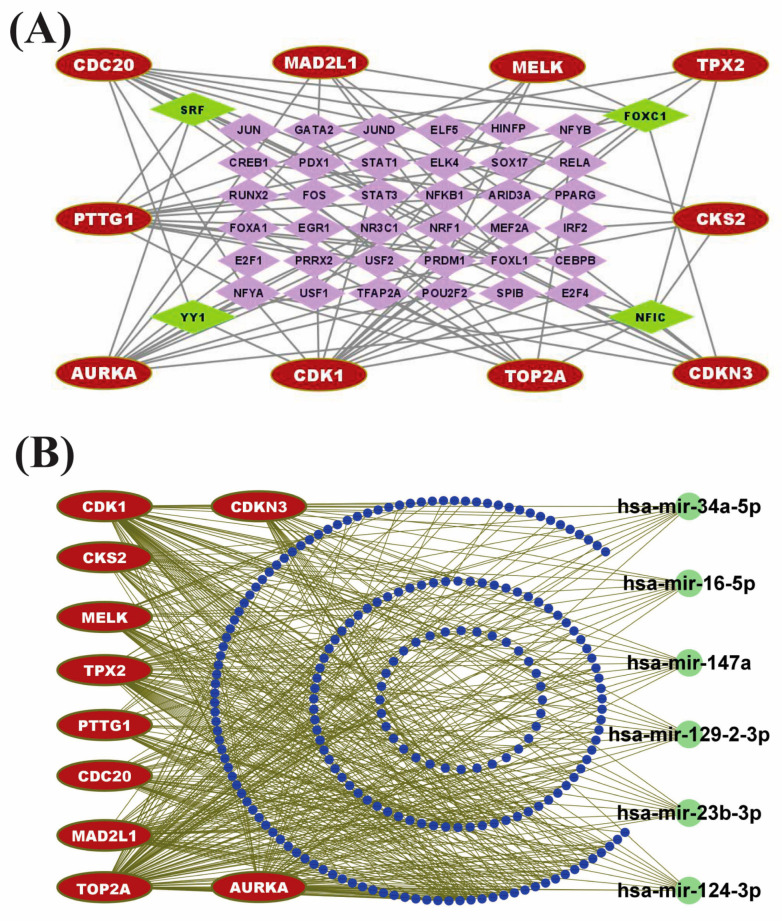
Regulatory network analysis of CGs. (**A**) TFs-CGs regulatory network contains 72 nodes and 174 edges. The red node indicates the core genes, whereas purple color nodes represent TFs. Among the TFs, green nodes represent the core TFs. (**B**) miRNA-CGs interaction network contains 223 nodes and 450 edges. The red node indicates the core genes, whereas blue nodes represent miRNAs. Among the miRNAs, green color represents the core miRNA.

**Figure 7 cancers-15-01369-f007:**
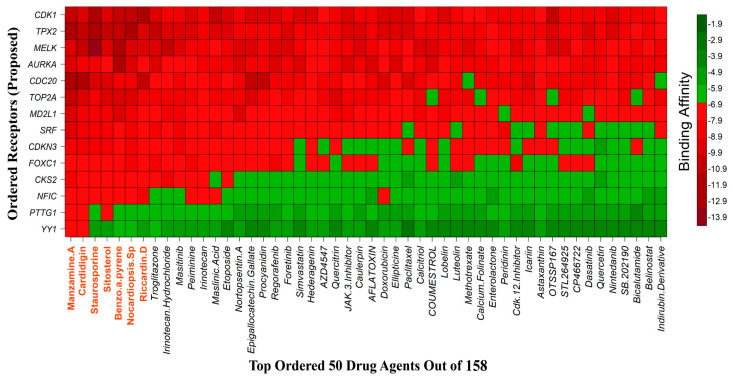
Matrix of molecular docking analysis results. The target-receptor proteins are ordered in a row and drug agents are ordered in a column, where red colors represent strong binding affinity. The red colors text in the row represents proposed drug agents.

**Figure 8 cancers-15-01369-f008:**
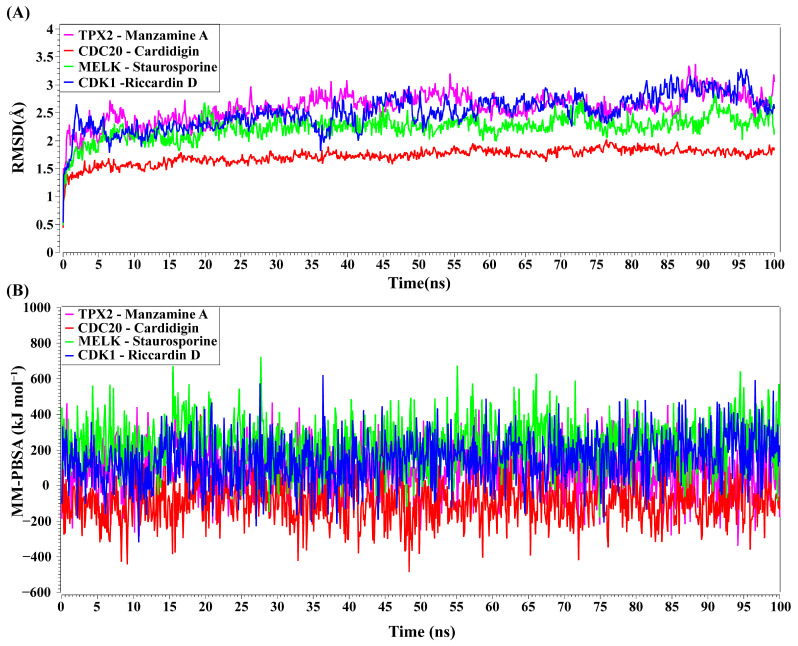
MD simulations of top-ranked complexes. (**A**) Time evolution of RMSDs for each of top-ranked complexes. (**B**) Binding stability of top-ranked four complexes by MM-PBSA binding free energy (kJ mol^−1^) against each ns of simulation; greater positive values show strong binding. Complexes: red color CDC20–Cardidigin, green color MELK–Staurosporine, pink color TPX2–Manzamine A, and blue color CDK1–Riccardin-D.

**Table 1 cancers-15-01369-t001:** Contingency table.

Predefined Gene-Set	CGs (Proposed)	Not CGs (Proposed)	Marginal Total
*i*th term of interest (*A_i_*)	*m_i_*	*M_i_ − m_i_*	*M_i_*
Complement of *A_i_* ($A_{i}^{c}$)	*t − m_i_*	*T − M_i_ − t + m_i_*	*T − M_i_*
Marginal total	*t*	*T − t*	*T* (Grand total)

**Table 2 cancers-15-01369-t002:** List of the top five significantly (*p*-value < 0.001) annotated GO terms and KEGG pathways by CGs.

GO ID	GO Term	*p*-Value	Associated CGs
Biological Process (BPs)			
GO:0044772	mitotic cell cycle phase transition	8.50 × 10^−10^	*MELK;CDK4;MYC;CDK1;AURKA;CDC25B;CDKN3*
GO:0031145	anaphase-promoting complex-dependent catabolic process	1.71 × 10^−8^	*CDC20;PTTG1;CDK1;AURKA;MAD2L1*
GO:0010389	regulation of G2/M transition of mitotic cell cycle	3.04 × 10^−7^	*TPX2;CDK4;CDK1;AURKA;CDC25B*
GO:0007052	mitotic spindle organization	3.94 × 10^−7^	*CDC20;TPX2;CENPN;AURKA;MAD2L1*
GO:0007346	regulation of mitotic cell cycle	7.34 × 10^−7^	*CDC20;CDK1;CKS2;CDC25B;MAD2L1*
**Molecular Function (MFs)**			
GO:0035173	histone kinase activity	2.65 × 10^−5^	*CDK1;AURKA*
GO:0008353	RNA polymerase II CTD heptapeptide repeat kinase activity	6.23 × 10^−5^	*CDK4;CDK1*
GO:0019901	protein kinase binding	1.15 × 10^−4^	*TOP2A;TPX2;CKS2;AURKA;CDC25B*
GO:0045236	CXCR chemokine receptor binding	1.28 × 10^−4^	*CXCL8;CXCL12*
GO:0097472	cyclin-dependent protein kinase activity	2.37 × 10^−4^	*CDK4;CDK1*
**Cellular Component (CCs)**			
GO:0005819	Spindle	1.07 × 10^−6^	*CDC20;TPX2;CDK1;AURKA;MAD2L1*
GO:0000307	cyclin-dependent protein kinase holoenzyme complex	3.41 × 10^−6^	*CDK4;CDK1;CKS2*
GO:1902554	serine/threonine protein kinase complex	6.50 × 10^−6^	*CDK4;CDK1;CKS2*
GO:0043232	intracellular non-membrane-bounded organelle	8.52 × 10^−5^	*TOP2A;CDC20;TPX2;CDK4;MYC;TRIP13;AURKA*
GO:0072686	mitotic spindle	2.23 × 10^−4^	*TPX2;CDK1;MAD2L1*
**Pathways**		***p*-Value**	**Associated CGs**
**KEGG Pathway**			
Cell cycle		2.15 × 10^−11^	*CDC20;PTTG1;CDK4;MYC;CDK1;CDC25B;MAD2L1*
Bladder cancer		7.19 × 10^−8^	*CXCL8;MMP1;CDK4;MYC*
Oocyte meiosis		1.48 × 10^−7^	*CDC20;PTTG1;CDK1;AURKA;MAD2L1*
Human T-cell leukemia virus 1 infection		2.04 × 10^−6^	*CDC20;PTTG1;CDK4;MYC;MAD2L1*
Progesterone-mediated oocyte maturation		2.68 × 10^−6^	*CDK1;AURKA;CDC25B;MAD2L1*

**Table 3 cancers-15-01369-t003:** The 1st, 2nd, 3rd column show potential targets, 2-dimentional(2d) structure of lead compounds, top ordered binding affinities (kcal mol^−1^), respectively. The 3-dimension(3d) view of top ranking drug-target complexes is shown in the 4th column. Finally, the last column shows key elements of interacting amino acids, including hydrogen bond, hydrophobic interactions, and electrostatic.

Potential Targets	Structure of Top Compounds	Binding Affinity Score (kcal mol^−1^)	3D Structures of Complex with Interactions	Interacting Amino Acids		
				Hydrogen Bond	Hydrophobic Interactions	Electrostatic
TPX2	Manzamine A 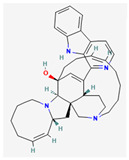	−12.4	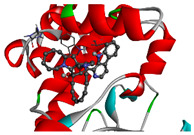	-	VAL317 TRP313 HIS366	-
CDC20	Cardidigin 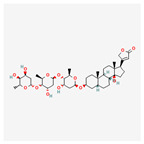	−11.0	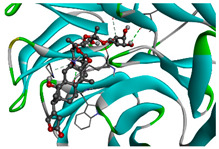	TRP317 PRO319 VAL190 LEU449 PRO319	LYS236	-
MELK	Staurosporine 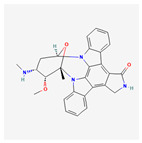	−13.4	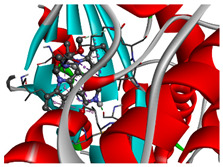	GLU87 GLU136 CYS89 ASP150	ILE17 VAL25 LEU139 LEU149 ALA38 ILE149 LYS40	-
CDK1	Riccardin D 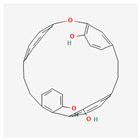	−11.3	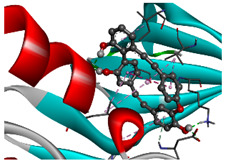	LEU83, ASP146, LYS33	VAL18 LEU135 ILE10 ALA31 VAL64	-

## Data Availability

All data revolved are describe in Section 2.1 (Data Sources and Descriptions).

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
