# Peer review of "Exploring Core Genes by Comparative Transcriptomics Analysis for Early Diagnosis, Prognosis, and Therapies of Colorectal Cancer"

_cancers, 2023, doi:10.3390/cancers15051369_

Round 1

Reviewer 1 Report

Cancers 2133624

Comments to authors:
The manuscript of Islam et al should be strongly revised in all its chapters tacking into accounts the following criticisms and suggestions

INTRODUCTION

lines88-89: the sentence should be rephrased since DEGs could NOT be considered as MUTATED genes

lines 91-98: 1) among the 9 cited references, 4 are from the same research group and two are not related to cancer; 2) The sentence “...so far, no researcher yet suggested a few core DEGs for early diagnosis, prognosis and therapies of CRC. “ is not true, since reference 13 is dedicated to “core
DEGs” in colorectal cancer” and reference 77 describes “identification of Hub genes...in colorectal cancer”

MATERIALS AND METHODS/RESULTS

Chapter 2.5 and Figure 3: it is necessary to better explain the comparison with “other diseases” because only cancers are included in the figure 4 and the two colors used in the figures seem to suggest an YES/NO expression without a scale of expected differences

RESULTS/DISCUSSION

Figure 3: it is necessary to comment that the identified core genes seem associated to many other cancers and in particular some of the genes seem not differentially expressed in Colon carcinoma (3 genes) and in colorectal cancer (2 genes). These data could also suggest a pancancer role of the
identified core genes.

Figure 4: is poorly commented and not siscussed (could be removed)

Figure 5: it is not clear which dataset was used for KM curves generation

Figure 6: COAD database is not cited in the materials and methods

DISCUSSION/CONCLUSIONS

Comparison and differences with already published paper (ref 13 and 77) are to be added.
Pros and cons of the choice of comparison normal/cancer and selection of only core genes should be presented and discussed. In fact this choice might introduce a selection for “cell cycle” and “mitotic spindle pathways” and could explain the similarity with many other types of cancer. These data indicate that no “tumor specific” diagnostic role could be attributed to the identified core genes. In addition the exclusion of analysis of the tumor progression at priori could explain the very limited differences among tumor stages (Figure 6). It is suggested to avoid the term validation in the conclusions. It is strongly suggested to chane/mitigate the last sentence of the conclusion.

Author Response

Response to Reviewer 1

The manuscript of Islam et al should be strongly revised in all its chapters tacking into accounts the following criticisms and suggestions

Response: Thanks for your important comments and suggestion. We tried our best to improve the quality of the manuscript according to your comments and suggestions.  

INTRODUCTION:

Comment (1): lines88-89: the sentence should be rephrased since DEGs could NOT be considered as MUTATED genes

Response: Thanks for your important comments. Yes, you are right. A DEGs may not always mutated.  So, we rewrote the sentences in the revised manuscript accordingly (see line 68-77).

Comment (2): lines 91-98: 1) among the 9 cited references, 4 are from the same research group, and two are not related to cancer;

Response: Thanks for your important query. We removed the irrelevant references in the revised manuscript (see line 78-80, ref [14-22]).

Comment (3):  The sentence “...so far, no researcher yet suggested a few core DEGs for early diagnosis, prognosis and therapies of CRC. “Is not true, since reference 13 is dedicated to “core DEGs” in colorectal cancer” and reference 77 describes “identification of Hub genes...in colorectal cancer”

Response: Thanks for the references. In the reference 13 (new ref 16), Xu et al. (2020) identified CRC-causing core DEGs and recommended their application for early diagnosis, prognosis and therapies of CRC. But they did not perform any analysis with their core DEGs in favor of early diagnosis and therapies. In the reference 77 (new ref. 22), Patil et al. (2021) identified CRC-causing core DEGs and recommended their application for early diagnosis of CRC. But they did not explore any drug molecules for therapies of CRC.

MATERIALS AND METHODS/RESULTS:

Comment (4): Chapter 2.5 and Figure 3: it is necessary to better explain the comparison with “other diseases” because only cancers are included in the figure 4 and the two colors used in the figures seem to suggest a YES/NO expression without a scale of expected differences.

Response: Thanks for your nice suggestions. Actually, we performed CGs-disease interaction analysis to verify the association between CGs (core genes) and CRC by the independent DisGeNET database which is constructed based on 21,671 genes and 30,170 diseases. The Enrichr web-tool measures the association of a disease with a group of CGs those are overlapped (common) with the reference genes-set of that disease. We selected top-ranked 20 diseases that are significantly enriched by the CGs with the cut-off Adj. p-value < 0.05.  We observed that all of those enriched diseases are different types of cancers including CRC and colon carcinoma only (please see Figure 5A in the revised manuscript). According to the setting of Enrichr web-tool, common (overlapping) genes between CGs-set and reference genes-set for a disease indicated red color, while uncommon (non-overlapping) CGs with those reference genes for that disease indicate green color. The Enrichr web-tool does not offer any pair-wise gene-disease association. So, the scaling of association is not suitable in this case.

RESULTS/DISCUSSION:

Comment (5): Figure 3: it is necessary to comment that the identified core genes seem associated to many other cancers and in particular some of the genes seem not differentially expressed in Colon carcinoma (3 genes) and in colorectal cancer (2 genes). These data could also suggest a Pan-cancer role of the identified core genes.  

Response: Thanks for your further suggestion regarding the same issue. We rewrote this section accordingly (Please see lines 271-283 and 411-412).    

Comment (6): Figure 4: is poorly commented and not discussed (could be removed)

Response: Thank you for your important comments. Figure 4 (new figure 6) indicates the gene-regulatory network (GRN) which provides the transcriptional and post-transcriptional regulatory factors of core genes (CGs). We discussed it clearly in the revised manuscript (see lines 174-184, 319-323, and 459-466)

Comment (7): Figure 5: it is not clear which dataset was used for KM curves generation.

Response: Thanks for the important query. Figure 5 (new figure 4B) uses TCGA-COAD and TCGA-READ database to construct multivariate KM survival probability curves based on the SurvExpress web tool. We added this information in the revised manuscript (see lines 134-136, 266-269).

Comment (8): Figure 6: COAD database is not cited in the materials and methods

Response: Thanks for your nice comments. We cited the database accordingly. (See lines 129-132)

DISCUSSION/CONCLUSIONS

Comment (9): Comparison and differences with already published paper (ref 13 and 77) are to be added. 

Response: Thanks for your nice suggestion. As early mentioned in the introduction section (Lines 78-87), In the reference 13 (ref. 16), Xu et al. 2020 identified CRC-causing hub-DEGs and recommended their application for early diagnosis, prognosis and therapies of CRC. But they that did not perform any analysis with their core DEGs in favor of early diagnosis and therapies. In the reference 77 (new ref. 22), Patil et al. (2021) identified CRC-causing 40 hub-DEGs for early diagnosis, which may be large number for further investigation by wet-lab experiment, since it would be laborious, time-consuming and costly. Also, they did not recommend any drug molecules for therapies of CRC. In our study, we proposed 10 hub-DEGs, where 9 genes were overlapped/common with their 40 hub-DEGs. In addition, we investigated their pan-cancer roles and recommended hub-DEGs guided 10 drug molecules for therapies of CRC. Therefore, the findings of our study might be more useful resources for early diagnosis, prognosis and therapies of CRC compare to the findings of Patil et al. 2021 (new ref. 22). We also added this statements in discussion and conclusion sections ( see Line 406-409) 

Comment (10): Pros and cons of the choice of comparison normal/cancer and selection of only core genes should be presented and discussed.

Response 1: Thanks for your important suggestion. Out of around 40,000 genes, we found 252 DEGs between CRC and control samples. Though the number of DEGs much smaller than the total number of genes, it may be yet large number for further investigation by wet-lab experiment, since it would be laborious, time-consuming and costly. Therefore, a smaller set of DEGs that will be the representative of all DEGs, is required for reducing time, cost and labor during the wet-lab experiment. Though total DEGs are more informative than any smaller set of DEGs, but the smaller representative set of DEGs would be more beneficial from the viewpoints of time, cost and labor. Therefore, we detected top-ranked 10 core/hub-DEGs for further investigation. We also added this statements in discussion and conclusion sections ( see Line 394-400) 

Comment (11): In fact, this choice might introduce a selection for “cell cycle” and “mitotic spindle pathways” and could explain the similarity with many other types of cancer.

Response: Thanks for your important suggestion again. We explained these issues in the revised manuscript (please see lines 426-445).

Comment (12): These data indicate that no “tumor-specific” diagnostic role could be attributed to the identified core genes.

Response: Thanks for your important comments. Figures 5A (disease-CGs association) and 5B & S2 (pan-cancer analysis) showed that our proposed core genes (CGs) may significantly play the diagnostic role for multiple cancers including CRC (please see lines 411-412).

Comment (13): In addition, the exclusion of analysis of the tumor progression at priori could explain the very limited differences among tumor stages (Figure 6). 

Response: Thanks for your important comments. Actually Figure 6 (new Figure 4A) shows that the average expressions of each core gene (CG) with the tumor progression are almost same in all stages of 1 to 4, but much larger from the normal stage, which indicates each core gene may play the diagnostic and prognostic role from the earlier stage of CRC (please see lines 271-276 and 403-406).   

Comment (14): It is suggested to avoid the term validation in the conclusions. It is strongly suggested to change/mitigate the last sentence of the conclusion.

Response: Thanks for your important suggestion. We rewrote this sentence accordingly (see lines 494-496) 

Reviewer 2 Report

This is an interesting manuscript where authors identified core genes associated with the prognosis and detection of CRCs. The manuscript is quite comprehensive, and the results are also interesting. I would also like to appreciate the authors for the wonderful analysis carried out by them.

Nonetheless, I have several remarks:

1.      There were many grammatical mistakes during the writing of this manuscript. Many of the sentences need to be reframed and checked for grammatical mistakes. A few of the sentences are:

“Therefore, the finding output of this study would be useful resource for early diagnosis, prognosis, and therapies of CRC”

“At first, we identified 252 cDEGs from the publicly available three gene 32 expression profiles with NCBI accession numbers”

“Therefore, the output of this study might be played a important role in early stages diagnosis, prognosis, and therapies for CRC”.

2.      How the current study is different from the previously cited studies in references 11-19 where differentially expressed genes (DEGs) have been identified in CRC patients’ tissue vs healthy colon?

3.      Can authors comment on the difference in the core genes that they found in this study in the distal vs proximal colon as these regions of the colon have different pathophysiology to cause CRC?

4.      Can authors comments If the findings of the current study can be functionally validated in the lab? If yes, then how? 

Author Response

Response to Reviewer 2

This is an interesting manuscript where authors identified core genes associated with the prognosis and detection of CRCs. The manuscript is quite comprehensive, and the results are also interesting. I would also like to appreciate the authors for the wonderful analysis carried out by them.

Nonetheless, I have several remarks:

Comment (1): There were many grammatical mistakes during the writing of this manuscript. Many of the sentences need to be reframed and checked for grammatical mistakes. A few of the sentences are:

“Therefore, the finding output of this study would be useful resource for early diagnosis, prognosis, and therapies of CRC”

“At first, we identified 252 cDEGs from the publicly available three gene expression profiles with NCBI accession numbers”

“Therefore, the output of this study might be played a important role in early stages diagnosis, prognosis, and therapies for CRC”.

Response: Thanks for your important comments and suggestion. We rewrote the above sentences. (Please, see lines, 26-27, 32-33, and 44-45). We also updated other sentences accordingly, where it is necessary. 

Comment (2):  How the current study is different from the previously cited studies in references 11-19 where differentially expressed genes (DEGs) have been identified in CRC patients’ tissue vs healthy colon?

Response: Thanks for your important comments. Actually, articles with references 11-18 (new ref 14-21), identified only CRC-causing core-DEGs highlighting their functions and pathways. They that did not perform any analysis with their core DEGs in favor of early diagnosis and therapies. The article with reference 19 (new ref 22), identified CRC-causing core-DEGs and suggested their application for early diagnosis of CRC. They identified CRC-causing 40 hub-DEGs for early diagnosis, which may be large number for further investigation by wet-lab experiment, since it would be laborious, time-consuming and costly. Also, they did not recommend any drug molecules for therapies of CRC. In our study, we proposed 10 hub-DEGs, where 9 genes were overlapped/common with their 40 hub-DEGs. In addition, we investigated their pan-cancer roles and recommended hub-DEGs guided 10 drug molecules for therapies of CRC. Therefore, the findings of our study might be more useful resources for early diagnosis, prognosis and therapies of CRC compare to the findings of Patil et al. 2021 (new ref. 22). We also discuss this issues in Lines 78-90, 406-409)

Comment (3): Can authors comment on the difference in the core genes that they found in this study in the distal vs proximal colon as these regions of the colon have different pathophysiology to cause CRC?

Response: Thanks for your impotent suggestions. Basically, we collected three gene expression profile datasets from NCBI-GEO database with accession ID GSE106582, GSE110223 and GSE74602, where there was no any description about distal or proximal colon regions. They just mentioned that the data was  collected from colon or rectal or colorectal regions, in general. 

Comment (4): Can authors comments If the findings of the current study can be functionally validated in the lab? If yes, then how? 

Response: Thank you very much for your important comment. In this study, we verified our findings by independents databases and literature review. It can be functionally validated by the in-vitro/qRT-qPCR or in-vivo approaches, but we have no any wet-lab facility. However, we hope that the findings of this study would be useful resources to the wet-lab researchers.

Round 2

Reviewer 1 Report

In the revised form the manuscript was strongly improved